# Impact of Continuous Flow Left Ventricular Assist Device on Heart Transplant Candidates: A Multi-State Survival Analysis

**DOI:** 10.3390/jcm11123425

**Published:** 2022-06-14

**Authors:** Massimiliano Carrozzini, Tomaso Bottio, Raphael Caraffa, Jonida Bejko, Olimpia Bifulco, Alvise Guariento, Carlo Mario Lombardi, Marco Metra, Danila Azzolina, Dario Gregori, Marny Fedrigo, Chiara Castellani, Vincenzo Tarzia, Giuseppe Toscano, Antonio Gambino, Vjola Jorgji, Enrico Ferrari, Annalisa Angelini, Gino Gerosa

**Affiliations:** 1Department of Cardiac Thoracic Vascular Sciences and Public Health, University of Padova, 35121 Padova, Italy; massimiliano.carrozzini@gmail.com (M.C.); raphael.caraffa@gmail.com (R.C.); jonidabg@gmail.com (J.B.); bif.oly92@gmail.com (O.B.); alvise.guariento@unipd.it (A.G.); danila.azzolina@unipd.it (D.A.); dario.gregori@unipd.it (D.G.); marny.fedrigo@gmail.com (M.F.); chiara.castellani@unipd.it (C.C.); v.tarzia@gmail.com (V.T.); giuseppe.toscano@aopd.veneto.it (G.T.); antonio.gambino@unipd.it (A.G.); annalisa.angelini@unipd.it (A.A.); gino.gerosa@unipd.it (G.G.); 2Cardiothoracic Unit, Spedali Civili of Brescia, Cardiology, Department of Medical and Surgical Specialties, Radiological Sciences and Public Health University, 25121 Brescia, Italy; lombardi.carlo@alice.it (C.M.L.); metramarco@libero.it (M.M.); 3Hacohen Lab, Massachusetts General Hospital, Boston, MA 02114, USA; vjorgji@broadinstitute.org; 4Cardiac Surgery Unit, Cardiocentro Ticino Institute, 6900 Lugano, Switzerland; enrico.ferrari@cardiocentro.org; 5Biomedicine Faculty, Italian Switzerland University (USI), 6900 Lugano, Switzerland; 6Cardiac Surgery Department, University Hospital of Zurich, University of Zurich, 8057 Zurich, Switzerland

**Keywords:** mechanical circulatory support heart transplant, left ventricle assist device, multi-state survival analysis, waitlist survival, bridge to transplant strategy

## Abstract

(1) Objectives: The aim of this study was to investigate the impact of the prolonged use of continuous-flow left ventricular assist devices (LVADs) on heart transplant (HTx) candidates. (2) Methods: Between January 2012 and December 2019, we included all consecutive patients diagnosed with end-stage heart failure considered for HTx at our institution, who were also eligible for LVAD therapy as a bridge to transplant (BTT). Patients were divided into two groups: those who received an LVAD as BTT (LVAD group) and those who were listed without durable support (No-LVAD group). (3) Results: A total of 250 patients were analyzed. Of these, 70 patients (28%) were directly implanted with an LVAD as BTT, 11 (4.4%) received delayed LVAD implantation, and 169 (67%) were never assisted with an implantable device. The mean follow-up time was 36 ± 29 months. In the multivariate analysis of survival before HTx, LVAD implantation showed a protective effect: LVAD vs. No-LVAD HR 0.01 (*p* < 0.01) and LVAD vs. LVAD delayed HR 0.13 (*p* = 0.02). Mortality and adverse events after HTx were similar between LVAD and No-LVAD (*p* = 0.65 and *p* = 0.39, respectively). The multi-state survival analysis showed a significantly higher probability of death for No-LVAD vs. LVAD patients with (*p* = 0.03) or without (*p* = 0.04) HTx. (4) Conclusions: The use of LVAD as a bridge to transplant was associated with an overall survival benefit, compared to patients listed without LVAD support.

## 1. Introduction

Heart failure (HF) is a critical high-risk condition of death, and its prevalence is expected to increase further [1,2,3]. Although heart transplantation (HTx) is the gold standard of therapies, its application is restricted by the limited number of suitable organ donors, and mortality among those on the waiting list remains high [4,5]. Recently, the use of continuous flow implantable left ventricular assist devices (LVADs) has transformed the management of patients waiting for HTx through a bridge-to-transplant (BTT) strategy [4,6,7,8]. Despite the satisfactory results of this mechanical support, comparative analyses versus the standard medical therapy in HTx candidates are still scant and not definitive [9,10]. Thus, the choice to implant a durable LVAD as BTT still remains largely dependent on clinicians’ and patients’ preference.

In this study, we sought to investigate the impact of durable LVAD as a BTT by examining the overall survival rate, the waiting list survival rate, and outcomes after HTx, compared to no-LVAD patients.

## 2. Materials and Methods

### 2.1. Ethics Statement

Every reasonable effort was made to obtain written informed consent to participate in this study. In particular, the use of data for scientific and research purposes has been included in the written informed consent agreements used. The local Institutional Review Board (University Hospital, Padua, Italy) approved the study design, consent process, data review, and analysis (IRB number 48401; 23 September 2021).

Anonymity, professional secrecy, and the use of the collected data were guaranteed. The statistical analyses were performed exclusively for the scientific purposes provided for by current legislation and in compliance with the ISHLT ethical declaration.

### 2.2. Study Design

A single-center retrospective analysis on prospectively collected data was performed. We reviewed all consecutive patients diagnosed with end-stage heart failure considered for HTx at our institution, between January 2012 and December 2019. We included in the study all patients who were both eligible for HTx and for an LVAD implant as BTT. The exclusion criteria were: LVAD implantation with a bridge-to-candidacy (BTC) strategy, ineligibility for LVAD implantation, and removal from the waiting list for reasons other than clinical worsening. Eligibility for HTx and LVAD therapy was concordant with the recommendations of the International Society for Heart and Lung Transplantation (ISHLT) [11].

Patients were divided into 2 groups: those who received an LVAD as a bridge to transplant (LVAD group) and those who were listed without durable support (No-LVAD group).

All patient characteristics, donor characteristics, intra- and peri-procedural HTx data, and follow-up data were collected through the review of medical records. Follow-up was completed in all patients. Primary outcomes were overall survival, survival before HTx, and post-HTx survival. Secondary outcomes were adverse events after HTx, including acute and chronic rejection.

### 2.3. LVAD Management

At our institution, all HTx candidates without contraindications for durable LVAD are recommended for LVAD implantation, following the last guidelines of the European Society of Cardiology, the European Association for Cardio-Thoracic Surgery, and ISHLT [1,12,13].

All patients were implanted with one of the following third generation LVAD devices: Jarvik 2000 FlowMaker ™ VAD (Jarvik Heart Inc, New York, NY, USA), HeartWare™ HVAD™ System (Medtronic, Dublin, Ireland), and HeartMate 3™ LVAD (Abbott, Chicago, IL, USA). The devices were implanted using standard procedures [10,14,15,16]. After implantation of the device, standard management and medical therapy followed [1,10,12,13,14,15,16].

### 2.4. Heart Transplant Management

All heart transplants were orthotopic and performed with a bicaval anastomosis. Standard induction immunosuppressive therapy was used [10,17,18]. Patients received regularly follow-ups in a dedicated clinic. Endomyocardial biopsies were performed at regular intervals: weekly for the first month (starting 15 days after HTx), twice weekly until the fourth month after HTx, and then monthly until the end of the first year. Thereafter, biopsies were repeated annually or in case of clinical suspicion [19,20,21].

### 2.5. Statistical Analysis

The normality of all continuous variables was tested using the Shapiro-Wilk test, and graphically assessed by histograms and Q–Q plots. Continuous variables are expressed as mean and standard deviation, while categorical variables are presented as number and percentage. The analysis of continuous variables was performed with a Student’s *t*-test while Pearson’s chi-square or Fisher’s exact tests (applied when one or more of the cell counts in the 2 × 2 table was less than 5) were used for categorical variables. Multivariate survival analysis was performed by the Cox regression risk model. In this model, covariate entries with a univariable *p*-value ≤ 0.20 were included and a conditional forward stepwise method was used. The proportionality of the risk was verified by means of the Grambsch and Therneau test [22]. Survival curves were plotted with the Kaplan–Meier method.

Transition frequencies and probabilities were reported for a 4-state outcome classification (No-LVAD, LVAD, HTx, death). An overall survival analysis of competitive risk was performed using a continuous time Markov multi-state transition model [23]. This model is particularly useful for longitudinal data as it describes a process in which a patient moves through a series of states in continuous time. The next state to which a patient moves, and the time of the change, is governed by a set of transition intensities $q_{ij}$ for each couple of states *i* and *j*. A transition matrix has been defined, whose rows add up to zero, so that the diagonal entries are defined by $q_{ij}=-\sum_{i\neq j}q_{ij}$. The transitions allowed between states were: No-LVAD to LVAD, No-LVAD to HTx, No-LVAD to death, LVAD to HTx, LVAD to death, and HTx to death. The probabilities of death in the follow-up times were graphed. The significance in the comparison of transition probabilities was calculated using a bootstrap analysis of 1000 runs.

Statistical analysis was performed with IBM (IBM, Armonk, NY, USA) © SPSS © Statistics 25 software and R 3.3.5 software with rms, survival and msm packages [23]. Significance was set at *p* < 0.05.

## 3. Results

### 3.1. Patient Population

A total of 315 patients matched the inclusion criteria. Among these, 65 patients were excluded (36 cases ineligible for LVAD, 20 deleted from the list for reasons other than clinical worsening, and 9 LVAD implants with BTC strategy), with a final study population of 250 patients. All included patients were HTx candidates, eligible for LVAD, for whom the device implantation was considered. The starting point for retrospective observation of patient outcomes was the listing date for HTx or the LVAD implant date.

Patient characteristics at baseline are shown in Table 1. Most patients (80%) were profiled as an INTERMACS class 3–4. Of the study population, 70 patients (28%) underwent direct LVAD implantation, while 180 (72%) were listed for HTx without durable mechanical circulatory support. In this second group, 11 patients underwent delayed LVAD implantation after a mean time of 6 ± 9 months. Across the entire study population, 190 patients (76%) achieved HTx: 55 in LVAD support (22%) and 135 (54%) without LVAD support.

### 3.2. Survival before HTx

In a mean time of 10 ± 12 months before HTx, a total of 30 (12%) deaths occurred. The impact of clinical characteristics and data at baseline on death before HTx was assessed by univariable and multivariable analyses (Table 1). The results showed No-LVAD, diabetes, and list status as independent predictors of death. The use of LVAD, with direct or delayed implantation, was protective: direct LVAD vs. No-LVAD hazard ratio (HR) = 0.01 (95% Confidence Interval (CI): 0.00–0.06; *p* < 0.01), and Delayed LVAD vs. no-LVAD HR = 0.13 (95% CI: 0.02–0.74; *p* = 0.02).

Overall mortality before HTx among direct LVAD patients was 6% (*n* = 4), whereas it was 36% (*n* = 4) in the delayed LVAD group and 13% (*n* = 22) in the No-LVAD group. K–M survival curves, stratified according to use of LVAD, were significantly divergent with the best survival estimate belonging to the direct LVAD patients group (Figure 1).

### 3.3. Outcomes after HTx

A total of 190 patients (76%) underwent heart transplantation: 55 (22%) with LVAD support and 135 (25%) without LVAD. Table 2 summarizes patient data at HTx. All baseline characteristics of Htx patients were comparable except for age (*p* = 0.019). Patients without LVAD were admitted more frequently to intensive care units (*p* < 0.01), on inotropic (*p* < 0.01) or temporary circulatory support (*p* < 0.01), with a higher rate of impaired renal function (*p* < 0.01) and lower cardiac index (*p* = 0.018). Donor characteristics were comparable between the two groups.

At a mean follow-up time of 42 ± 30 months for the LVAD and 34 ± 29 months for the group without LVAD (*p* = 0.08), the mortality rate was comparable in the two groups (16% LVAD vs. 19% No-LVAD, *p* = 0.64). K–M survival curves were similar (Figure 1). Although cardiopulmonary bypass time was longer in LVAD cases (*p* < 0.01), the adverse event rate after HTx was comparable between the two groups (Table 3). Effusions (*p* < 0.01) and tracheostomy (*p* = 0.015) occurred more often in patients without LVAD who, on average, spent a longer period in intensive care (*p* = 0.01).

### 3.4. Acute and Chronic Rejection Rate

The allo-sensitization rate before HTx was comparable between patients with LVAD and without LVAD (Table 3). After HTx, acute cell rejection rates were similar, considering any degree of rejection and only the highest degrees (≥2R). Calculation of the cell rejection score (CRS) at 3, 6 and 12 months also produced comparable results.

Donor specific antibodies (DSAs) were detected in a higher percentage of patients with LVAD, although not significantly. On the other hand, the antibody-mediated rejection rate was significantly higher in the No-LVAD group than in LVAD (11% vs. 0%, *p* = 0.01). Accordingly, chronic allograft vasculopathy appeared to develop more frequently in cases without LVAD, but this finding did not reach significance.

### 3.5. Overall Survival Analysis

Considering the overall period (i.e., both pre and post HTx), the mortality rate was 19% (*n* = 13) for direct LVAD, 36% (*n* = 4) for delayed LVAD and 29% (*n* = 48) for No-LVAD patients. The KM survival curves (Figure 1) showed significantly higher survival for LVAD patients than for those without LVAD (*p* = 0.02).

In addition to these results, a multi-state model was applied for the analysis of overall survival in order to overcome the potential bias related to the analysis of concurrent events. Our model reproduced all possible treatment pathways, as shown in Figure 2. The three possible treatments (no-LVAD, LVAD and HTx) and death are the states allowed by the model. The result of the analysis is a transition probability matrix. We plotted the probability of transit, over time, from the different possible treatments to death (Figure 2). With further calculations (1000 bootstrap replicas), statistical significance for the different state changes was derived (Table 4). Multi-state model analysis showed a significantly higher probability of death for patients without LVAD than for those with LVAD with (*p* = 0.03) or without (*p* = 0.04) HTx.

## 4. Discussion

In the current era, there is a growing gap between the number of advanced heart failure patients waiting for an HTx and the limited availability of donor organs [24,25]. In this context, continuous-flow implantable LVADs could represent a life-saving therapy. Generally, only randomized clinical trials are recommended to establish a degree of evidence suitable for changing practice. Currently, randomized trials on LVAD therapy focusing on BTT strategy are lacking. The main reason for this is that it remains impossibile to randomize an out-patient HTx candidate to a double surgical step (LVAD and HTx) versus direct HTx. In our study, we focused on this promising strategy and designed a single-center retrospective analysis of HTx candidates suitable for LVAD implantation, dividing them by the pre-transplant strategy used (third-generation continuous-flow LVAD implant as BTT vs. no LVAD implant). Results were evaluated using traditional statistics and a multi-state model to compete for survival risk analysis.

The use of LVAD significantly improved survival before HTx. In addition to this, our multi-state model showed an overall significant survival benefit in LVAD patients. INTERMACS registry data showed that 1 year survival after implantation increased from 65% to 81% with continuous flow pumps [8]. However, a clear advantage of using LVAD for survival in HTx candidates has not yet been demonstrated. A possible approach to the question is to frame the observation through the pre- and post-HTx survival times, considering them separately.

Comparative studies on the impact of LVAD in the period before HTx are scarce in the literature. Thagavi et al. compared the results of pulsatile-LVAD vs. no-LVAD vs. CF-LVADs (2nd and 3rd generation). They showed significantly improved survival before HTx with continuous-flow devices compared to the other two groups, although multivariate analysis failed to demonstrate any association with improved survival [26]. Trivedi and colleagues analyzed data from the United Network for Organ Sharing (UNOS) database and showed that the 1 year survival of waiting list patients connected to HeartMate II was significantly higher than in patients without LVAD [27]. A more recent analysis of the UNOS registry confirmed these results [28].

In our cohort, survival analysis of the period prior to HTx showed that patients assisted with LVAD had a significantly higher expected survival rate, while, on the other hand, implantation times greatly influenced outcomes, since delayed implantation was associated with higher mortality. Patients with delayed implantation were patients who underwent LVAD treatment only after a progressive decline in their clinical condition. This result is perfectly in line with the international literature [6,7,8]. In addition, we can speculate that a rigorous outpatient follow-up (performed in our center on LVAD patients) is of fundamental importance for the outcomes of LVAD implantation and for pre-Htx survival rates. One could argue that patients with LVAD support have a prolonged waiting time before HTx. As a matter of fact, according to Italy’s national allocation system, a patient with uncomplicated LVAD support does not have high urgency status. However, two considerations should be made: first, HTx from a high urgency status could prevent the most appropriate donor-recipient matching, possibly affecting the outcomes; second, as for our institutional policy, we aim to transplant LVAD patients within the first 2 years of support, to reduce the impact of device-related adverse events.

In terms of post-Htx outcomes, we observed comparable results in patients with and without LVAD in our cohort. Despite this finding, we interestingly observed a higher rate of clinical worsening at Htx for cases without LVAD. This may mean that HTx is able to mitigate the potentially harmful effects of waiting without durable mechanical circulatory support. The most recent analysis of the ISHLT heart transplant registry did not support our results and showed that the use of LVAD is a significant risk factor for both 1 year post-HTx mortality (HR 1.42, *p* < 0.01) and at 5 years (1.34, *p* < 0.01) [29]. However, this analysis did not specify the type of LVAD considered, potentially including intra- and para-corporeal mixed generation devices.

Despite the interesting considerations drawn from the separate analysis of the pre- and post-HTx results, we sought to broadly assess the overall impact of LVADs. Overall survival analysis showed significantly higher survival in patients with LVAD than in those without LVAD. Nevertheless, this approach is potentially limited by the presence of several concurrent treatments and events. To overcome these limitations, we have applied a multi-state survival model that provides a reliable assessment of the processes in which subjects move through different states over time. In our study, the possible states were No-LVAD, LVAD, HTx, and death. By analyzing the probabilities of transition between the different treatments and death, we were able to assess the risk of mortality associated with each. The results showed a significantly higher mortality rate in patients without LVAD than in cases of LVAD. This difference was maintained regardless of the transition to HTx. On this basis, we could characterize the overall effect of using LVAD as a protective strategy. Based on the post-HTx outcomes discussed above, we can further hypothesize that the risk reduction associated with durable mechanical circulatory support is due to the greater likelihood that patients with LVAD will achieve HTx.

We then evaluated the immunological implications of LVAD implantation, a topic of particular debate in the literature [30]. In our population, allo-sensitization recorded at the pre-HTx time was found to be balanced between patients with or without durable mechanical support. This trend remained the same even after HTx, where acute cell rejection rates were similar regardless of LVAD use, even when calculating CRS at different time points. Conversely, we found a higher incidence of antibody-mediated rejection in No-LVAD cases. This peculiar finding could be related to the considerable proportion of No-LVAD cases undergoing short-term mechanical support, which may have increased their immunological risk. Further subgroup analyses are needed to better clarify this result.

## 5. Limitations

This study has some limitations. First, it is a retrospective review of a cohort of patients that spans nearly a decade, and, therefore, includes different surgical techniques, postoperative management, and devices. In addition to this, despite our encouraging efforts in pursuing LVAD therapy, some patients refused sustained mechanical circulatory support, resulting in an unbalanced sample size in the groups analyzed.

## 6. Conclusions

In conclusion, our study demonstrates that the use of LVAD as a BTT is associated with a significantly reduced mortality risk in HTx candidates. Durable mechanical support was protective before HTx, while results after it were comparable to those in unassisted patients. LVAD implantation in elective patients carries no significant immunological impact.

## Figures and Tables

**Figure 1 jcm-11-03425-f001:**
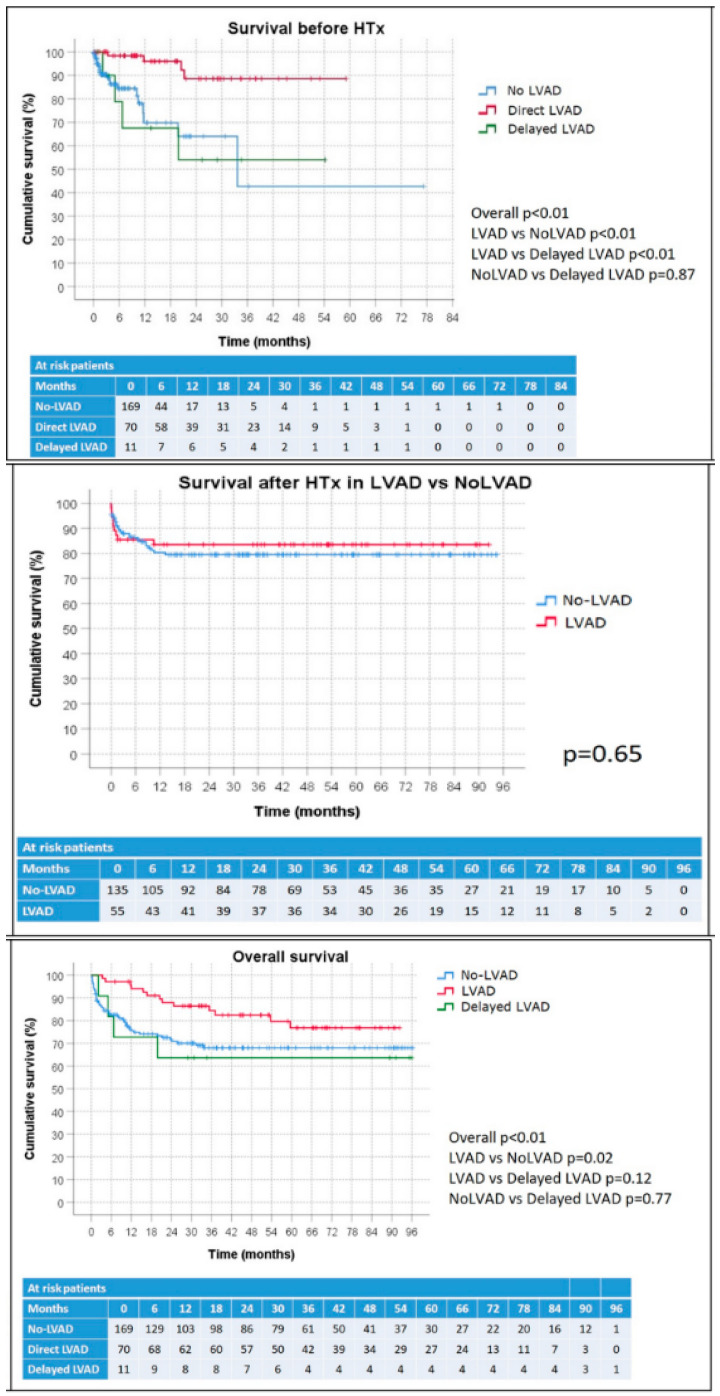
Kaplan–Meier plots of survival before HTx stratified according to LVAD use, survival after HTx in LVAD vs. no-LVAD patients, and overall survival according to LVAD use.

**Figure 2 jcm-11-03425-f002:**
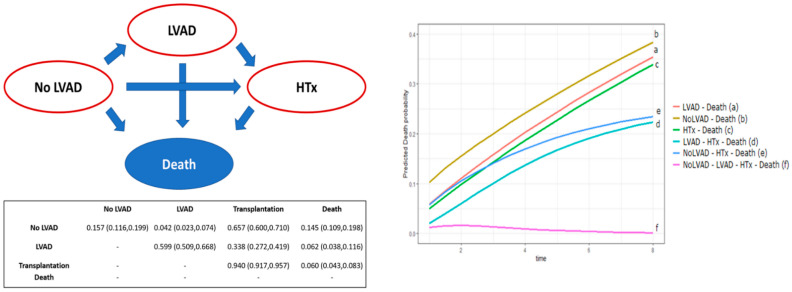
Multi-state model for survival analysis with transition probability (and 95% confidence interval) matrix and curves of death probabilities by treatment.

**Table 1 jcm-11-03425-t001:** Baseline patient’s data and univariable and multivariable analysis of death predictor.

	Univariable Analysis		Multivariable Analysis			
	*n* (%) or Mean ± SD	*p* Value	HR	Lower CL	Upper CL	*p* Value
Treatment received		0.01				<0.01
No-LVAD	169 (68%)		Direct LVAD vs. No-LVAD: 0.010	0.002	0.057	<0.01
Direct LVAD	70 (28%)		Delayed LVAD vs. No-LVAD: 0.132	0.024	0.738	0.021
Delayed LVAD	1 (4%)					
Age (years)	54 ± 13	0.820				
Female sex	51 (20%)	0.809				
BSA (m^2^)	1.8 ± 0.2	0.661				
Blood group		0.013				
A	101 (41%)					
B	28 (11%)					
AB	8 (3%)					
0	110 (45%)					
Cardiac diagnosis		0.599				
Primary dilated	9 (39%)					
Ischemic	109 (44%)					
Other cardiomyopathy	44 (18%)					
Acute onset	48 (19%)	0.036				
Dyslipidemia	95 (38%)	0.149				
Hypertension	95 (38%)	0.173				
Diabetes	55 (22%)	0.011	2.930	1.149	7.470	0.024
Peripheral artery disease	15 (6%)	0.512				
COPD	25 (10%)	0.517				
ICD/CRTD	202 (81%)	0.707				
Previous cerebral event	43 (17%)	0.343				
Previous smoking	94 (38%)	0.188				
INTERMACS profile						
1	45 (18%)	0.049				
2	6 (2%)					
3	52 (21%)					
4	147 (59%)					
Waiting list status (at entry)		0.079				<0.01
2B	120 (48%)		2Avs2B: 12.439	12,439	3160	<0.01
2A	97 (39%)		1vs2B: 1241.170	1,241,170	60,164	<0.01
1	5 (2%)		HUvs2B: 52.953	52,953	8419	<0.01
HU	28 (11%)					
Peak-VO_2_ (mL/kg/min)	11.5 ± 2.7	0.760				
Cardiac index (L/min/m^2^)	2.1 ± 0.6	0.875				
SPAP (mmHg)	40.4 ± 15	0.998				
PVR (WU)	2.3 ± 2.0	0.792				
PLTs (×10^9^/L)	216 ± 76	0.917				
Bilirubin (µmol/L)	21 ± 14	0.148				
C-reactive protein (mg/L)	24 ± 42	<0.01				
GFR (mL/min/m^2^)	68 ± 26	0.112				
ICU stay	81 (32%)	0.343				
Inotropic infusion	95 (42%)	0.877				
Mechanical ventilation	7 (3%)	<0.01				
CRRT	10 (4%)	<0.01				
IABP	5 (2%)	0.493				
Temporary MCS	49 (20%)	<0.01				
Time before HTx (months)	10 ± 12	0.266				

BSA, body surface area. GFR, glomerular filtration rate. COPD, chronic obstructive pulmonary disease. CRRT, continuous renal replacement therapy. IABP, intra-aortic balloon pump. ICU, intensive care unit. ICD/CRTD, implantable cardioverter defibrillator/cardiac resynchronization therapy defibrillator. INTERMACS, Interagency Registry for Mechanically Assisted Circulatory Support. MCS, Mechanically Circulatory Support. PVR, pulmonary vascular resistance. SPAP, systolic pulmonary artery pressure. PLTs, platelets.

**Table 2 jcm-11-03425-t002:** Patient characteristics at HTx and donor data.

	All Patients (*n* = 190)	No-LVAD (*n* = 135)	LVAD (*n* = 55)	*p*-Value
Age	54 ± 14	56 ± 13	50 ± 16	0.019
Female sex	43 (23%)	35 (26%)	8 (15%)	0.089
BSA (m^2^)	1.8 ± 0.2	2 ± 0	1.8 ± 0.5	
Blood group				0.643
A	83 (44%)	59 (44%)	24 (45%)	
B	27 (15%)	22 (16%)	5 (9%)	
AB	8 (4%)	6 (4%)	2 (4%)	
0	70 (37%)	48 (36%)	22 (42%)	
Cardiac diagnosis				0.062
Primary dilated	75 (40%)	50 (37%)	25 (46%)	
Ischemic	82 (43%)	56 (42%)	26 (47%)	
Other cardiomyopathy	33 (17%)	29 (22%)	4 (7%)	
Dyslipidemia	67 (35%)	49 (36%)	18 (33%)	0.641
Hypertension	72 (38%)	51 (38%)	21 (38%)	0.958
Diabetes	31 (16%)	21 (16%)	10 (18%)	0.657
Peripheral artery disease	11 (6%)	9 (7%)	2 (4%)	0.515
COPD	19 (10%)	14 (10%)	5 (9%)	0.790
ICD/CRTD	148 (78%)	109 (81%)	39 (71%)	0.139
Previous cerebral event	35 (18%)	25 (19%)	10 (18%)	0.957
Previous smoking	69 (36%)	45 (33%)	24 (44%)	0.180
Last waiting list status				<0.01
2B	74 (39%)	74 (55%)	0 (0%)	
2A	60 (32%)	17 (13%)	43 (78%)	
1	10 (5%)	7 (5%)	3 (6%)	
HU	46 (24%)	37 (27%)	9 (16%)	
Peak-VO_2_ (mL/kg/min)	11.7 ± 2.4	12.2 ± 2.5	11.8 ± 2.3	0.560
Cardiac index (L/min/m^2^)	2.1 ± 0.6	2.5 ± 2.8	2.0 ± 1.0	0.018
SPAP (mmHg)	37 ± 15	39 ± 16	33 ± 11	0.010
PLTs (×10^9^/L)	211 ± 83	207 ± 86	221 ± 74	0.275
Bilirubin (µmol/L)	25 ± 38	37 ± 58	23 ± 26	0.330
C-reactive protein (mg/L)	38 ± 66	68 ± 85	45 ± 61	0.250
GFR (ml/min/m^2^)	70 ± 32	63 ± 30	84 ± 30	<0.01
ICU stay	61 (32%)	51 (38%)	10 (18%)	<0.01
Inotropic infusion	61 (32%)	54 (41%)	7 (13%)	<0.01
Mechanical ventilation	7 (4%)	6 (5%)	1 (2%)	0.676
CRRT	9 (5%)	9 (7%)	0 (0%)	0.061
IABP	1 (1%)	1 (1%)	0 (0%)	1.000
Temporary MCS	38 (20%)	37 (27%)	1 (2%)	<0.01
Time before HTx (months)	8 ± 10	4 ± 6	16 ± 14	<0.01
Donor age	45 ± 17	46 ± 16	43 ± 19	0.210
Female to male	34 (25%)	22 (23%)	12 (27%)	0.623
Donor inotropic infusion	132 (71%)	91 (70%)	41 (76%)	0.377
Donor cardiac arrest	37 (20%)	25 (19%)	12 (22%)	0.688
Donor smoker	47 (25%)	34 (26%)	13 (24%)	0.740
Donor diabetes	4 (2%)	2 (2%)	2 (4%)	0.583
Donor cold ischemia time	204 ± 59	202 ± 60	209 ± 56	0.483

BSA, body surface area. GFR, glomerular filtration rate. COPD, chronic obstructive pulmonary disease. CRRT, continuous renal replacement therapy. IABP, intra-aortic balloon pump. ICU, intensive care unit. ICD/CRTD, implantable cardioverter defibrillator/ cardiac resynchronization therapy defibrillator. MCS, Mechanically Circulatory Support. SPAP, systolic pulmonary artery pressure. PLTs, platelets.

**Table 3 jcm-11-03425-t003:** Outcomes after HTx.

	**All Patients** **(*n* = 190)**	**No-LVAD** **(*n* = 135)**	**LVAD** **(*n* = 55)**	***p*-Value**
Cardiopulmonary bypass time (min)	209 ± 64	200 ± 62	228 ± 67	<0.01
ECMO	44 (23%)	29 (22%)	15 (27%)	0.420
Mechanical ventilation (hours)	83 ± 171	92 ± 187	62 ± 124	0.282
CRRT	67 (35%)	51 (38%)	16 (29%)	0.242
Documented Infection	65 (35%)	48 (36%)	17 (31%)	0.497
Severe bleeding	23 (12%)	20 (15%)	3 (6%)	0.087
Cerebral event	22 (12%)	15 (11%)	17 (13%)	0.765
Ischemic stroke	17 (9%)	11 (8%)	6 (11%)	0.556
Hemorrhagic stroke	3 (2%)	2 (2%)	1 (2%)	1.000
Bowel ischemia	10 (5%)	6 (5%)	4 (7%)	0.482
Major arrhythmia	11 (6%)	9 (7%)	2 (4%)	0.513
Pericardial/pleural effusions	39 (21%)	35 (26%)	4 (7%)	<0.01
Tracheostomy	19 (10%)	18 (13%)	1 (2%)	0.015
Hepatic or pancreatic complication	13 (7%)	11 (8%)	2 (4%)	0.352
ICU stay (days)	10 ± 13	11 ± 14	7 ± 6	0.01
Hospital stay (days)	42 ± 26	43 ± 29	38 ± 18	0.132
Follow-up time (months)	36 ± 29	34 ± 29	42 ± 30	0.08
30-day death	16 (9%)	10 (8%)	6 (11%)	0.449
In-hospital death	18 (13%)	12 (13%)	6 (13%)	0.926
Overall post-HTx death	35 (18%)	26 (19%)	9 (16%)	0.641
**Immunological findings**				
	**All patients** **(*n* = 190)**	**No-LVAD** **(*n* = 135)**	**LVAD** **(*n* = 55)**	***p* Value**
Baseline positive CDC	9 (5%)	7 (5%)	2 (4%)	1.000
Baseline positive CDC >10%	3 (2%)	3 (2%)	0 (0%)	0.558
Baseline positive Luminex class 1	23 (12%)	18 (13%)	5 (9%)	0.472
Baseline positive Luminex class 2	15 (8%)	14 (10%)	1 (2%)	0.07
Baseline positive Luminex class 1/2	30 (16%)	24 (18%)	6 (11%)	0.232
Donor specific antibody	17 (16%)	9 (12%)	8 (24%)	0.108
Acute cellular rejection (any grade)	153 (89%)	107 (88%)	46 (90%)	0.736
Acute cellular rejection (≥grade 2R)	66 (38%)	50 (41%)	16 (31%)	0.220
Cellular rejection score 3 months	0.46 ± 0.35	0.44 ± 0.36	0.50 ± 0.33	0.328
Cellular rejection score 6 months	0.47 ± 0.30	0.46 ± 0.31	0.51 ± 0.27	0.373
Cellular rejection score 12 months	0.46 ± 0.26	0.45 ± 0.28	0.48 ± 0.22	0.622
Antibody mediated rejection	13 (8%)	13 (11%)	0 (0%)	0.011
Chronic allograft vasculopathy	3 (2%)	3 (4%)	0 (0%)	0.276

ECMO, extracorporeal membrane oxygenation. CRRT, continuous renal replacement therapy. ICU, intensive care unit.

**Table 4 jcm-11-03425-t004:** *p* values for comparison among transition probabilities in the multi-state model.

	LVAD–Death	No-LVAD–Death	HTx–Death	LVAD–HTx–Death	No-LVAD–HTx–Death	No-LVAD–LVAD–HTx–Death
LVAD–Death	-	0.038	0.217	0.032	0.221	0.006
No-LVAD–Death	-	-	0.022	<0.001	0.033	<0.001
HTx–Death	-	-	-	0.059	0.207	0.018
LVAD–HTx–Death	-	-	-	-	0.026	0.338
No-LVAD–HTx–Death	-	-	-	-	-	0.008

## Data Availability

Data supporting the findings of this study are available upon request from the corresponding author [TB]. The data are not publicly available due to restrictions, since these informations could compromise the privacy of research participants.

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
