# Peer review of "Impact of Continuous Flow Left Ventricular Assist Device on Heart Transplant Candidates: A Multi-State Survival Analysis"

_jcm, 2022, doi:10.3390/jcm11123425_

Round 1

Reviewer 1 Report

Carrozzini at al sought to investigate the impact of prolonged use of continuous-flow left ventricular assist devices (LVADs) on heart transplant (HTx) patients. The authors found that the use of LVAD as a bridge for transplantation was associated with a significantly reduced mortality risk after HTx, compared to patients without LVAD support.

In the concluding sentence is a logic mistake. In fact pre-HTx implantation of LVAD reduced mortality rate before HTx and therefore also overall mortality in patients with LVAD, but LVAD implantation before HTx did not reduce mortality rate after HTx. This kind of mistake please correct in the whole paper.

Plase provide sampe size calculation.

30 patients died before HTx, 190 were transplanted. What about the remaining 30?

Where there any differences in survival before and after HTx in relation to the type of LVAD?

Author Response

 Reviewer Comments and Suggestions for Authors

Reviewer 1
Comment 1

Carrozzini at al sought to investigate the impact of prolonged use of
continuous-flow left ventricular assist devices (LVADs) on heart transplant
(HTx) patients. The authors found that the use of LVAD as a bridge for
transplantation was associated with a significantly reduced mortality risk
after HTx, compared to patients without LVAD support.
In the concluding sentence is a logic mistake. In fact pre-HTx implantation
of LVAD reduced mortality rate before HTx and therefore also overall
mortality in patients with LVAD, but LVAD implantation before HTx did not
reduce mortality rate after HTx. This kind of mistake please correct in the
whole paper.

Answer 1: we agree with the reviewer, this point was clarified in the whole paper.

Comment 2: Plase provide sampe size calculation.
Answer 2: we did not calculated the sample size as this was a retrospective analysis

Comment 3: 30 patients died before HTx, 190 were transplanted. What about the remaining 30?
Answer 3: remaining patients were alive still waiting for transplant.

Comment 4: Where there any differences in survival before and after HTx in relation to the type of LVAD?
Answer 4: we did not investigated possible differences among different types of LVAD used because this was not the aim of our study. We compared LVAD versus no-LVAD patients.

Reviewer 2

Comment 1: Carrozzini et al. analyzed the impact of continuous flow left ventricular assist device on heart transplant survival
Major: - Please discuss the dilemma of patients loosing their high urgency
   status after LVAD implantation (for EuroTransplant at least). It this the
   case for the authors' country likewise?
Answer 1: according to our national allocation system, a patient with uncomplicated LVAD support have not a high urgency status. This could lead to a prolonged waiting time before HTx. However, two considerations should be made: first, HTx from a high urgency status could prevent from the most appropriate donor-recipient matching, possibly affecting the outcomes; second, as for our institutional policy, we aim to transplant LVAD patients within the first 2 years of support.

These considerations were included in the discussion (lines 267-273).

Comment 2: The authors state "All consecutive patients diagnosed with end-stage heart failure and receiving an HTx at the University of Padua between January 2012 and December 2019 were included." Please include patients dying or otherwise being excluded while waiting for a heart transplantation. Otherwise, this might lead to a major bias in my humble opinion. All patients listed for heart transplantation should be included.

Answer 2: we agree with the reviewer, this was a misleading statement. As a matter of fact, we reviewed for inclusion in the study all consecutive patients diagnosed with end-stage heart failure considered for HTx at our Institution. Thus, not only transplanted patients were included.

This point was clarified throughout the whole paper.

Comment 3: The comparison of patients with LVAD vs. patients with no LVAD might be prone for bias. Can the authors clearly describe the criteria when a patients receive an LVAD in their institution? One could speculate, patients with early LVAD implantation might have different clinical characteristics. Importantly, the patients in the cohort without LVAD are older, have a higher sPAP and a lower GFR at time of transplantation. Can the authors present baseline characteristics for the three groups (no LVAD, LVAD early, LVAD late) at time of HTx listing?

Answer 3: at our Institution, all HTx candidates without contraindications for durable LVAD are proposed for LVAD implant. This point was included in the methods (line 121-122).

Waiting list survival before HTx was analysed in the pooled study population by a Cox regression, using durable LVAD as a covariate. This was to account for potential confounding factors at the baseline.

Comment 4: For a sounder analysis, the authors might match patients with LVAD to patients without LVAD implantation by propensity score to dimish bias.

Answer 4: in our study we dealt with different treatments, through which a given patient could pass, and competing events. In this statistical scenario, we believe that the provided multi-state survival analysis is more appropriate than a propensity score matching.

Comment 5: Please indicate/explain on which occasion chi-squared test or Fisher's exact test was used.

Answer 5: the Fisher’s exact test was applied when one or more of the cell counts in the 2×2 table was less than 5. This was clarified in the methods(lines 143-144).

Comment 6:
Minor:
   - please correct: last Author "Gino Gerosa ->and<- MD"
   - please update the ESC guideline reference to the 2021 version
   - Please rephrase, difficult to understand: "The reason for this lies in
   the psycho-social-clinical difficulty of accepting by a 55 patient
   (INTERMACS IV) the chance of being transplanted according to a single-step
   56 strategy (direct HTx) and / or a double surgical step (LVAD-HTx)"
   - Introduction l. 47: "limit" used twice in a sentence
   - please use "cleaner" graphics (for example Figure 1)
Answer 6: all these minor revisions were made in the paper

Reviewer 2 Report

Carrozzini et al. analyzed the impact of continuous flow left ventricular assist device on heart transplant survival

Major:

  • Please discuss the dilemma of patients loosing their high urgency status after LVAD implantation (for EuroTransplant at least). It this the case for the authors' country likewise?
  • The authors state "All consecutive patients diagnosed with end-stage heart failure and receiving an HTx at the University of Padua between January 2012 and December 2019 were included." Please include patients dying or otherwise being excluded while waiting for a heart transplantation. Otherwise, this might lead to a major bias in my humble opinion. All patients listed for heart transplantation should be included.
  • The comparison of patients with LVAD vs. patients with no LVAD might be prone for bias. Can the authors clearly describe the criteria when a patients receive an LVAD in their institution? One could speculate, patients with early LVAD implantation might have different clinical characteristics. Importantly, the patients in the cohort without LVAD are older, have a higher sPAP and a lower GFR at time of transplantation. Can the authors present baseline characteristics for the three groups (no LVAD, LVAD early, LVAD late) at time of HTx listing?
  • For a sounder analysis, the authors might match patients with LVAD to patients without LVAD implantation by propensity score to dimish bias.
  • Please indicate/explain on which occasion chi-squared test or Fisher's exact test was used.

Minor:

  • please correct: last Author "Gino Gerosa ->and<- MD"
  • please update the ESC guideline reference to the 2021 version
  • Please rephrase, difficult to understand: "The reason for this lies in the psycho-social-clinical difficulty of accepting by a 55 patient (INTERMACS IV) the chance of being transplanted according to a single-step 56 strategy (direct HTx) and / or a double surgical step (LVAD-HTx)"
  • Introduction l. 47: "limit" used twice in a sentence
  • please use "cleaner" graphics (for example Figure 1)

Author Response

(The authors gave the same response as above.)
